# Accurate Neural Training with 4-bit Matrix Multiplications at Standard Formats

**Brian Chmiel**[†∘]   **Ron Banner**[†]   **Elad Hoffer**[†]   **Hilla Ben Yaacov**[†]   **Daniel Soudry**[∘]

[†]Habana Labs – An Intel company, Caesarea, Israel,
[∘]Department of Electrical Engineering - Technion, Haifa, Israel

```
{bchmiel, rbanner, ehoffer, hbyaacov}@habana.ai
{daniel.soudry}@gmail.com
```

## Abstract

Quantization of the weights and activations is one of the main methods to reduce the computational footprint of Deep Neural Networks (DNNs) training. Current methods enable 4-bit quantization of the forward phase. However, this constitutes only a third of the training process. Reducing the computational footprint of the entire training process requires the quantization of the neural gradients, i.e., the loss gradients with respect to the outputs of intermediate neural layers.

Previous works separately showed that accurate 4-bit quantization of the neural gradients needs to (1) be unbiased and (2) have a log scale. However, no previous work aimed to combine both ideas, as we do in this work. Specifically, we examine the importance of having unbiased quantization in quantized neural network training, where to maintain it, and how to combine it with logarithmic quantization. Based on this, we suggest a *logarithmic unbiased quantization* (LUQ) method to quantize both the forward and backward phases to 4-bit, achieving state-of-the-art results in 4-bit training without the overhead. For example, in ResNet50 on ImageNet, we achieved a degradation of 1.1%. We further improve this to a degradation of only 0.32% after three epochs of high precision fine-tuning, combined with a variance reduction method—where both these methods add overhead comparable to previously suggested methods. A reference implementation is supplied in the supplementary material.

## 1 Introduction

Deep neural networks (DNNs) training consists of three main general-matrix-multiply (GEMM) phases: the forward phase, backward phase, and update phase. Quantization has become one of the main methods to compress DNNs and reduce the GEMM computational resources. Previous works showed the weights and activations in the forward pass to 4 bits while preserving model accuracy (Banner et al., 2019; Nahshan et al., 2019; Bhalgat et al., 2020; Choi et al., 2018b). Despite these advances, they only apply to a third of the training process, while the backward phase and update phase are still computed with higher precision.

Recently, Sun et al. (2020) was able, for the first time, to train a DNN while reducing the numerical precision of most of its parts to 4 bits with some degradation (e.g., 2.49% error in ResNet50). To do so, Sun et al. (2020) suggested a non-standard radix-4 floating-point format, combined with double quantization of the neural gradients (called two-phase rounding). This was an impressive step forward in the ability to quantize all GEMMs in training. However, since a radix-4 format is not aligned with conventional radix-2, any numerical conversion between the two requires an explicit multiplication to modify both the exponent and mantissa. Thus, their non-standard quantization requires specific hardware support (Kupriianova et al., 2013) that can significantly reduce the benefit of quantization to low bits (Appendix A.6), and make it less practical.

The main challenge in reducing the numerical precision of the entire training process is quantizing the neural gradients, i.e. the backpropagated error. Previous works showed separately that, to achieve accurate low precision representation of the neural gradients, it is important to use: (1) Logarithmic

quantization and (2) Unbiased quantization. Specifically, Chmiel et al. (2021) showed the neural gradients have a heavy tailed near-lognormal distribution and found an analytical expression for the optimal floating point format. At low precision levels, the optimal format is logarithmically quantized. For example, for FP4 the optimal format is [sign,exponent,mantissa] = [1,3,0], i.e. without mantissa bits. In contrast, weights and activations are well approximated with Normal or Laplacian distributions (Banner et al., 2019; Choi et al., 2018a), and therefore are better approximated using uniform quantization (e.g., INT4). However, Chmiel et al. (2021) did not use unbiased quantization (nor did any of the previous works that use logarithmic quantization of the neural gradients (Li et al., 2020; Miyashita et al., 2016; Ortiz et al., 2018)). Therefore, they were unable to successfully quantize in this FP4 format (their narrowest format was FP5).

Chen et al. (2020a) showed that unbiased quantization of the neural gradients is essential to get unbiasedness in the weight gradients, which is required in SGD analysis convergence (Bottou et al., 2018). However, they focused on quantization using integer formats, as did other works that pointed out on the importance of being unbiased (Banner et al., 2018; Zhong et al., 2022). Naive quantization of the neural gradients using the optimal FP4 format (logarithmic) results in biased estimates of the FP32 weight gradients—and this leads to severe degradation in the test accuracy. For example, a major issue is that under aggressive (naive) quantization many neural gradients with magnitudes below the representable range are zeroed, resulting in biased estimates of the FP32 gradients and reduced model accuracy.

Using either a logarithmic scale or unbiased rounding alone catastrophically fails at 4bit quantization of the neural gradients (e.g., see Fig. 2 below). Therefore, it is critical to combine them, as we do in this paper in Section 4. To do this, we stochastically quantize gradients below the representable range to either zero or the smallest representable magnitude $\alpha$ to provide unbiased estimates within that "underflow" range. Additionally, in order to represent the maximum magnitude without bias, we dynamically adjust $\alpha$ so that the maximum can always be represented with an exponentiated scaling starting at $\alpha$. Finally, to completely eliminate bias, we devise an efficient way to use stochastic rounding on a logarithmic scale, on the values between $\alpha$ and the maximum. Together, this gradient quantization method is called *Logarithmic Unbiased Quantization (LUQ)*, and for 4-bit quantization it uses a numerical format with one sign bit, three exponent bits, and zero mantissa bits, along with stochastic mapping (to zero or $\alpha$) of gradients whose values are below $\alpha$ and stochastic rounding within the representable range.

**Main contribution** LUQ, for the first time, combines logarithmic quantization with unbiased quantization for the neural gradients and does this efficiently using a standard format. By additionally quantizing the forward phase to INT4, we enable, for the first time, an efficient scheme for "full 4-bit training", i.e. the weights, activations and neural gradients are quantized to 4-bit in standard formats (see Appendix A.1) so all GEMMs can be done in 4-bit, and also bandwidth can be reduced. As we show, this method requires little to no overhead while achieving state-of-the-art accuracy results: for example, in ResNet50 we get $1.1\%$ error degradation with standard formats; in comparison, the previous method (Sun et al., 2020) had $2.49\%$ error degradation but required non-standard formats, as well as additional modifications which have additional overhead.

Moreover, in Section 5 we suggest two optional simple methods to further reduce the degradation, with some overhead: the first method reduces the quantization variance of the neural gradients using re-sampling, while the second is fine-tuning in high precision. Combining LUQ with these two proposed methods we achieve, for the first time, only $0.32\%$ error in the 4-bit training of ResNet50. The overhead of our additional methods is no more than similar modifications previously suggested in Sun et al. (2020). Lastly, in Section 7 we discuss how to reduce remaining overheads such as data movement, scaling operations, and GEMM-related operations.

## 2 RELATED WORKS

Neural networks Quantization has been extensively investigated in the last few years. Most of the quantization research has focused on reducing the numerical precision of the weights and activations for inference (e.g., Courbariaux et al. (2016); Rastegari et al. (2016); Banner et al. (2019); Nahshan et al. (2019); Choi et al. (2018b); Bhalgat et al. (2020); Choi et al. (2018a); Liang et al. (2021)). In this case, for standard ImageNet models, the best performing methods can achieve quantization to 4 bits with small or no degradation Choi et al. (2018a); Sakr et al. (2022). These methods can

be used to reduce the computational resources in approximately a third of the training (Eq. (1)). However, without quantizing the neural gradients, we cannot reduce the computational resources in the remaining two thirds of the training process (Eq. (2) and Eq. (3)). An orthogonal approach is low precision for the gradients of the weights in distributed training (Alistarh et al., 2016; Bernstein et al., 2018) in order to reduce the bandwidth and not the training computational resources.

Sakr & Shanbhag (2019) suggest a systematic approach to design a full training using fixed point quantization which includes mixed-precision quantization. Banner et al. (2018) first showed that it is possible to use INT8 quantization for the weights, activations, and neural gradients, thus reducing the computational footprint of most parts of the training process. Concurrently, Wang et al. (2018) was the first work to achieve full training in FP8 format. Additionally, they suggested a method to reduce the accumulator precision from 32bit to 16 bits, by using chunk-based accumulation and floating point stochastic rounding. Later, Wiedemann et al. (2020) showed full training in INT8 with improved convergence, by applying a stochastic quantization scheme to the neural gradients called non-subtractive-dithering (NSD). Also, Sun et al. (2019) presented a novel hybrid format for full training in FP8, while the weights and activations are quantized to [1,4,3] format, the neural gradients are quantized to [1,5,2] format to catch a wider dynamic range. Fournarakis & Nagel (2021) suggested a method to reduce the data traffic during the calculation of the quantization range, using the moving average of the tensor's statistics.

While it appears that it is possible to quantize to 8-bits all computational elements in the training process, 4-bits quantization of the neural gradients is still challenging. Chmiel et al. (2021) suggested that this difficulty stems from the heavy-tailed distribution of the neural gradients, which can be approximated with a lognormal distribution. This distribution is more challenging to quantize in comparison to the normal distribution which is usually used to approximate the weights or activations (Banner et al., 2019). Different works (Li et al., 2020; Miyashita et al., 2016) tried to use logarithmic quantization for the neural gradients, however, they failed to quantize them unbiasedly.

Sun et al. (2020) was the first work that presented a method to reduce the numerical precision to 4-bits for the vast majority of the computations needed during DNNs training. They use known methods to quantize the forward phase to INT4 and suggested quantizing the neural gradients twice with a non-standard radix-4 FP4 format. The use of the radix-4, instead of the commonly used radix-2 format, allows for covering a wider dynamic range. The main problem with their method is the specific hardware support for their suggested radix-4 datatype, which may limit the practicality of implementing their suggested data type.

Chen et al. (2020b) suggested reducing the variance in neural gradients quantization by dividing them into several blocks and quantizing each to INT4 separately. They require expensive sorting. Additionally, their per-sample quantization do not allow the use of an efficient GEMM operation.

## 3 BACKGROUND: QUANTIZATION FORMATS AND ROUNDING SCHEMES

Which quantization schemes should we use in 4bit training? Previous works (Choi et al., 2018a; Banner et al., 2019) showed that weights and activations can be quantized to INT4 with little to no accuracy degradation. In contrast, for the neural gradients, a recent work (Chmiel et al., 2021) showed analytically that the optimal format is logarithmic ([1,3,0]). Combining all these schemes, we focus on full 4-bit training using standard formats, with the following three 4-bit quantized GEMMs:

$$\textbf{[Forward]} \quad z_l = Q_{\text{INT}}(W_l)Q_{\text{INT}}(a_{l-1}); \qquad a_l = f_l(z_l) \tag{1}$$

$$\textbf{[Backward]} \quad g_{l-1} = Q_{\text{INT}}(W_l^T)Q_{\text{FP}}(\delta_l); \quad \delta_l = f_l'(z_l) \odot g_l \tag{2}$$

$$\textbf{[Update]} \quad \frac{\partial C}{\partial W_l} = Q_{\text{FP}}(\delta_l)Q_{\text{INT}}(a_{l-1}^T), \tag{3}$$

where $C$ is the loss function, $\odot$ is a component-wise product and, in each layer $l$, $f_l$ is the activation function, the weights ($W_l$) and activations ($a_l$) are quantized with INT4 ($Q_{\text{INT}}$) while the neural gradients $\delta_l \triangleq \frac{\partial C}{\partial z_l}$ are quantized with logarithmic FP4 ($Q_{\text{FP}}$), $z_l$ are the pre-activations, and $g_l \triangleq \frac{\partial C}{\partial a_l}$.

Next, we aim to find which rounding scheme should we use in each quantizer. Thus, we study the effects of unbiased rounding during the three phases of training (Eqs. 1, 2, and 3). We show that rounding-to-nearest (RDN) should be applied for the weights and activations ($Q_{\text{INT}}$), while the unbiased method of stochastic rounding (SR) is more suitable for the neural gradients ($Q_{\text{FP}}$).

### 3.1 MEAN SQUARE ERROR COMPARISON

In this section, we show that, although stochastic rounding (SR) is unbiased, it generically has a worse mean-square error (MSE) compared to round-to-nearest (RDN). Given that we want to quantize $x$ in a bin with a lower limit $l(x)$ and an upper limit $u(x)$, stochastic rounding can be stated as follows:

$$\mathrm{SR}(x) = \begin{cases} l(x), & w.p. \quad p(x) = 1 - \frac{x - l(x)}{u(x) - l(x)} \\ u(x), & w.p. \quad 1 - p(x) = \frac{x - l(x)}{u(x) - l(x)} \end{cases}. \tag{4}$$

The expected rounding value is given by

$$E[\mathrm{SR}(x)] = l(x) \cdot p(x) + u(x) \cdot (1 - p(x)) = x, \tag{5}$$

where here and below the expectation is over the randomness of SR (i.e., $x$ is a deterministic constant). In Table 1 we present the bias, variance, and MSE of RDN and SR. Full derivatives appear in Appendix A.2.

Table 1: Comparison of the bias, variance, and MSE of two different rounding schemes: round-to-nearest (RDN) and stochastic rounding (SR)

| Rounding | Bias | Variance | MSE |
|----------|------|----------|-----|
| RDN | $\min\left(x - l(x), u(x) - x\right)$ | $0$ | $\left[\min\left(x - l(x), u(x) - x\right)\right]^2$ |
| SR | $0$ | $(x - l(x)) \cdot (u(x) - x)$ | $(x - l(x)) \cdot (u(x) - x)$ |

From Table 1, since $\min(a, b)^2 \le a \cdot b$ for every $a, b$, we have that[1]

$$\mathrm{MSE}\left[SR\left(x\right)\right] \ge \mathrm{MSE}[RDN(x)], \quad \forall x. \tag{6}$$

In Fig. 1a we plot the mean-square-error for $x \in [0, 1]$, $l(x) = 0$, and $u(x) = 1$. However, while round-to-nearest has a lower MSE than SR, the former is a biased estimator.

### 3.2 WHEN IS IT IMPORTANT TO USE UNBIASED QUANTIZATION?

To prove convergence, textbook analyses of SGD typically assume the expectation of the (mini-batch) weight gradients is sufficiently close to the true (full-batch) gradient (e.g., assumption 4.3 in (Bottou et al., 2018)). This assumption is satisfied when the weight gradients are unbiased. Next, we explain (as pointed out by previous works, such as Chen et al. (2020a)) that the weight gradients are unbiased when the neural gradients are quantized stochastically without bias.

Recall $W_l$ and $f_l$ are, respectively, the weights and activation function at layer $l$ and $C$ is the cost function. Given an input–output pair $(x, y)$, the loss is:

$$C\left(y, f_L\left(W_L f_{L-1}\left(W_{L-1} \cdots f_2\left(W_2 f_1\left(W_1 x\right)\right) \cdots\right)\right)\right). \tag{7}$$

**Backward and Update Phases** Recall $z_l$ and $a_l$ are, respectively, the pre- and post- activations of layer $l$, and $\delta_l \triangleq \frac{dC}{dz_l}$. Defining $\delta_l^q \triangleq Q_{\mathrm{FP}}(\delta_l)$, we show in Appendix A.3 that the gradient of the weights in layer $l$ is $\nabla_{W_l} C = \delta_l a_{l-1}^\top$ and its quantized form is $\nabla_{W_l} C_q = \delta_l^q a_{l-1}^\top$. Therefore, the update $\nabla_{W_l} C_q$ is an unbiased estimator of $\nabla_{W_l} C$:

$$E\left[\nabla_{W_l} C_q\right] = E\left[\delta_l^q a_{l-1}^\top\right] = E\left[\delta_l^q\right] a_{l-1}^\top = \delta_l a_{l-1}^\top = \nabla_{W_l} C. \tag{8}$$

**Forward phase** The forward phase is different from the backward and updates phases in that unbiasedness at the tensor level is not necessarily a guarantee of unbiasedness at the model level since the activation functions and loss functions are not linear. For example, suppose we have two weight layers $W_1, W_2$, activation $f$, input $x$, and an SR quantizer $Q$. Then, despite that $Q$ is unbiased (i.e., $EQ(x) = x$), we get:

$$\mathbb{E}[f(W_2 Q(f(W_1 x)))] \ne \mathbb{E}[f(W_2(f(W_1 x)))] \tag{9}$$

since $f$ is non-linear. This means there is no point to use SR in the forward path since it will increase the MSE (Eq. (6)), but it will not fix the bias issue.

---

[1]Note Eq. (6) implies that $\int p(x)\mathrm{MSE}[SR(x)]dx \ge \int p(x)\mathrm{MSE}[RDN(x)]dx$ for any distribution $p(x)$.

### 3.3 Conclusions: When to use each rounding scheme?

Following the above results, the activation and weights quantization in the forward phase ($Q_{INT}$ in eq. Eq. (1)) should use RDN. This is because SR will increase the MSE (as shown in Eq. (6)), an increase which typically harms the final accuracy[2], but will not help make the loss estimate unbiased, due to the non-linearity of the loss and activation functions (e.g., Eq. (9)). To avoid mismatch (and additional bias), we use RDN in $Q_{\text{INT}}$ also in the other phases of training.

As we explained in section 3.2, unbiased neural gradients quantization leads to an unbiased estimate of the weight gradients, which enables proper convergence of SGD (Bottou et al., 2018). Thus, bias in the gradients can hurt the performance and should be avoided, even at the cost of increasing the MSE. Therefore, neural gradient quantization ($Q_{\text{FP}}$), should be done using a SR rounding scheme, following subsection 3.2. In Figs. 1b and 1c we see that these theoretical observations are consistent with empirical observations favoring RDN for the weights and activations ($Q_{\text{INT}}$) and SR for the neural gradients ($Q_{\text{FP}}$).

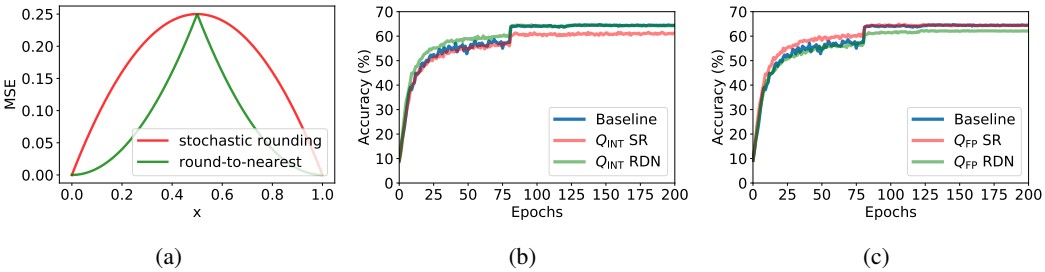

(a)  (b)  (c)

Figure 1: Comparison between stochastic rounding (SR) and round-to-nearest (RDN) quantization. In **(a)** we present the MSE of a uniform distributed tensor with the two different rounding schemes. Quantization to 4 bits of the activations and weights ($Q_{\text{INT}}$) **(b)** and neural gradients ($Q_{\text{FP}}$) **(c)** of ResNet18 - Cifar100 dataset with SR and RDN. While MSE is important in $Q_{\text{INT}}$ for the weights and activations, unbiasedness achieved with SR is crucial for the neural gradients in $Q_{\text{FP}}$. The neural gradients in (b) and the weights and activations in (c), are in full precision to focus on the effect of the rounding scheme only in one part of the network in each experiment.

## 4 LUQ: A LOGARITHMIC UNBIASED QUANTIZER

Following the conclusions from the previous section, the neural gradients quantizer ($Q_{\text{FP}}$ in eqs. 2 and 3 ) should be logarithmic and be completely unbiased. In this section we aim to do this efficiently and create, for the first time, a Logarithmic Unbiased Quantizer (LUQ).

Standard radix-2 floating-point defines a dynamic range. In standard FP, all the values below the minimum FP representation are pruned to 0 and all the values above the maximum FP representation are clipped to the maximum. In order to create a fully unbiased quantizer, we need to keep all the following three regions unbiased: below range minimum, inside range, and above range maximum.

**1) Below FP minimum: Stochastic underflow** Given an underflow threshold $\alpha$ we define a stochastic pruning operator, which prunes a given value $x$, as

$$T_\alpha(x) = \begin{cases} x & \text{, if } |x| \geq \alpha \\ \text{sign}(x) \cdot \alpha & w.p. \ \frac{|x|}{\alpha}, \text{if } |x| < \alpha \\ 0 & w.p. \ 1 - \frac{|x|}{\alpha}, \text{if } |x| < \alpha. \end{cases} \tag{10}$$

**2) Above FP maximum: Underflow threshold** In order to create an unbiased quantizer, the largest quantization value $2^{2^{b-1}}\alpha$ should avoid clipping any values in a tensor $x$, otherwise this will create a bias. Therefore, the maximal quantization value is chosen as $\max(|x|)$, the minimal value which will avoid clipping and bias. Accordingly, the underflow threshold $\alpha$ is (with $b = 3$ for FP4)

$$\alpha = \frac{\max(|x|)}{2^{2^{b-1}}}. \tag{11}$$

---

[2]There are some cases where adding limited noise, such as dropout, locally increases the MSE but improves generalization. However, this is typically not the case, especially if the noise is large.

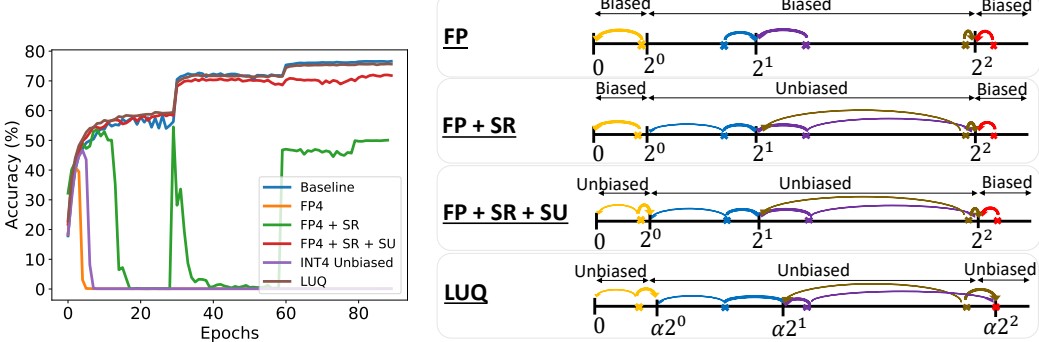

Figure 2: **(Left):** ResNet50 top-1 validation accuracy in ImageNet dataset with different quantization schemes for the neural gradients. FP4 refers to standard logarithmic (1-3-0) floating point quantization. SR refers to stochastic rounding, which makes the quantization unbiased inside the FP range. SU refers to stochastic underflow which makes the quantization unbiased below minimum FP. "INT4 unbiased" refers to the combination of INT4 and SR which is fully unbiased. Notice that while biased logarithmic quantization ("FP4"), partially biased logarithmic ("FP4 + SR", "FP4 + SR + SU"), and uniform unbiased ("INT4 unbiased") lead to significant accuracy degradation, the proposed fully unbiased logarithmic quantization ("LUQ") has a minimal degradation. **(Right):** Illustration of the different logarithmic quantization schemes for FP2 ([1,1,0] format), where $2^0$ and $2^2$ are, respectively, the minimal and maximal FP representations. Two arrows for the same point mean SR - thicker lines represent higher probability. Only LUQ is able to achieve unbiasedness in all floating point ranges.

The operation in Eq. (11) can have some overhead, but in Section 7 we present two methods ("Reducing the data movement" and "Reducing the cost of the scaling operation") to decrease this overhead.

**3) Inside FP range: Logarithmic SR**   Given an underflow threshold $\alpha$, let $Q_\alpha(x)$ be a FP round-to-nearest $b$-bits quantizer with bins $\{\alpha, 2\alpha, ..., 2^{2^{b-1}}\alpha\}$. Assume, without loss of generality, $2^{n-1}\alpha < x < 2^n\alpha$ ($n \in \{0, 1..., b-1\}$) . We will use the following quantizer, which is a special case of SR (Eq. (4)), and is unbiased as a special case of Eq. (5).:

$$Q_\alpha(x) = \begin{cases} 2^{n-1}\alpha & w.p.\ \ \frac{2^n\alpha - x}{2^n\alpha - 2^{n-1}\alpha} \\ 2^n\alpha & w.p.\ \ 1 - \frac{2^n\alpha - x}{2^n\alpha - 2^{n-1}\alpha} = \frac{x - 2^{n-1}\alpha}{2^{n-1}\alpha} \ . \end{cases} \tag{12}$$

The naive implementation of stochastic rounding can be expensive since it cannot use the standard quantizer. Traditionally, in order to use the standard quantizer the implementation includes simply adding uniform random noise $\epsilon \sim U[-\frac{2^{n-1}\alpha}{2}, \frac{2^{n-1}\alpha}{2}]$ to $x$ and then using a round-to-nearest operation. The overhead of such stochastic rounding is typically negligible (Appendix A.5.1) in comparison to other operations in neural networks training. Moreover, it is possible to reduce any such overhead with the re-use of the random samples (Appendix A.5.2). In our case, in order to implement a logarithmic round-to-nearest, we need to correct an inherent bias since $\alpha \cdot 2^{\lfloor \log(\frac{|x|}{\alpha})\rceil} \neq \alpha \cdot \lfloor 2^{\log(\frac{|x|}{\alpha})}\rceil$.

For a bin $[2^{n-1}, 2^n]$, the midpoint $x_m$ is

$$x_m = \frac{2^n + 2^{n-1}}{2} = \frac{3}{4} \cdot 2^{n-1} \ . \tag{13}$$

Therefore, we can apply round-to-nearest-power (RDNP) directly on the exponent $x$ of any value $2^{n-1} \leq 2^x \leq 2^n$ as follows:

$$\text{RDNP}(2^x) = 2^{\lfloor \log\left(\frac{4}{3} \cdot 2^x\right)\rfloor} = 2^{\lfloor x + \log\left(\frac{4}{3}\right)\rfloor} = 2^{\text{RDN}\left(x + \log\left(\frac{4}{3}\right) - \frac{1}{2}\right)} \approx 2^{\text{RDN}(x - 0.084)} \ . \tag{14}$$

Notice that the use of RDNP avoids the need of converting back to linear space in order to implement SR and avoid additional computational overhead.

**Logarithmic unbiased quantization (LUQ)**   LUQ, the quantization method we suggested above, is unbiased since it can be thought of as applying logarithmic stochastic rounding (Eq. (12)) on top of stochastic pruning (Eq. (17))

$$X_q = Q_\alpha\left(T_\alpha(x)\right) \ . \tag{15}$$

Since $T_\alpha$ and $Q_\alpha$ are unbiased, $X_q$ is an unbiased estimator for $x$, from the law of total expectation,

$$E[X_q] = E\left[Q_\alpha\left(T_\alpha(x)\right)\right] = E\left[E\left[Q_\alpha\left(T_\alpha(x)\right)\right]|T_\alpha(x)\right] = E\left[T_\alpha(x)\right] = x\,, \tag{16}$$

where the expectation is over the randomness of $T_\alpha$ and $Q_\alpha$.

In Fig. 2 (Left) we show an ablation study of the effect of the different quantization schemes on ResNet50 in ImageNet: while standard (biased) FP4 diverges, adding stochastic rounding or stochastic underflow (which make the process partially unbiased) enables convergence, but with significant degradation. Combining logarithmic quantization with full unbiasedness in LUQ obtained minimal accuracy degradation. Notice also that only unbiasedness without logarithmic quantization ("INT4 unbias") completely diverges. In Fig. 2 (Right) we show an illustration of the different logarithmic quantization schemes, where only LUQ achieved a fully logarithmic unbiasedness FP quantization.

## 5 Optional methods

Next, we present two optional methods to improve accuracy at some computational cost.

### 5.1 SMP: Reducing the variance while keeping it unbiased

In the previous section, we presented an unbiased method for logarithmic quantization of the neural gradients called LUQ. Following the bias-variance decomposition, if the gradients are now unbiased, then the only remaining issue should be their variance. Therefore, we suggest an optional method to reduce the quantization variance by repeatedly sampling from the stochastic quantizers in LUQ, and averaging the resulting samples of the final weight gradients. The proposed sampling can be implemented serially or in parallel. The serial implementation (re-using the same quantizer) has a power and throughput overhead but does not requires additional hardware (area) support, so it should be used if the chip area is the bottleneck. The parallel implementation avoids almost completely the throughput overhead (except the averaging operation), but it requires additional area for the multiple quantizers, so it should be used when the throughput is the bottleneck. For $N$ different samples, the proposed method will reduce the variance by a factor of $\frac{1}{N}$, without affecting the bias (Gilli et al., 2019). In Appendix Fig. 5 we show the effect of the different number of samples (SMP) on 2-bit quantization of ResNet18 Cifar100 dataset. There, with 16 samples, we achieve accuracy similar to a full-precision network. This demonstrates that the variance is the only remaining issue in neural gradient quantization using LUQ and that the proposed averaging method can erase this variance gap, with some overhead.

A different approach to reduce the variance can be to increase the bitwidth in the update phase. In order to keep using standard formats, we should increase the update phase to FP8-INT4, which leads to $3.5\times$ degradation in compute density in comparison to FP4-INT4 as shown in Sun et al. (2020) (Table s-1). Therefore, using the proposed SMP with two samples (as we shall do) has a significant advantage in compute density ($\mathbf{1.75\times}$) in comparison to increasing the bitwidth.

### 5.2 FNT: Fine-tuning in high precision

After the 4-bit training is finished, we suggest an optional method to reduce the gap from the full precision model, by running $T$ additional iterations in which we increase all the network parts to higher precision, except the weights which remain in low precision. We noticed that with this scheme we get the best accuracy for the fine-tuned network. At inference time the activations and weights are quantized to lower precision. During the fine-tune phase, the Learning Rate (LR) is increased linearly during $\frac{T}{2}$ iterations and then reduced linearly with the same slope:

$$\text{LR}_t = \begin{cases} \text{LR}_T + \frac{(\text{LR}_{\text{base}} - \text{LR}_T)}{T/2} & \text{, if } t \le \frac{T}{2} \\ \text{LR}_T \cdot \frac{(T-t)}{T/2} & \text{, else} \end{cases}\,, \tag{17}$$

where $\text{LR}_T$ is the final LR of the 4-bit training and $\text{LR}_{\text{base}}$ is the maximal LR of the fine-tune phase.

## 6 Experiments

In this section, we evaluate the proposed LUQ for 4-bit training on various DNN models. For all models, we use their default architecture, hyper-parameters, and optimizers combined with a

custom-modified Pytorch framework that implemented all the low precision schemes. Additional experimental details appear in Appendix A.4.

**INT4 quantization**   INT4 quantization methods for the weights and activations (forward pass) were well studied in the past. In this paper, we used SAWB Choi et al. (2018a) to quantize the weights and activations. SAWB determines the quantization scaling factor by first finding the optimal (in terms of MSE) scaling factor on six distribution approximations of the true tensor distribution, and then applying linear regression to find the chosen scaling factor.

**Training time measurement**   Notice that, currently, AI accelerators do not support 4-bit formats for training. This means that we can only simulate the quantization process, but are not able to measure training time or memory reduction. This is the common practice in the neural network quantization literature, where the algorithms often appear before the hardware that can support them. For example, though we can find FP8 training publications since 2019 (Sun et al., 2019), only recently did Nvidia announce their first GPU that supports the FP8 format (H100).

**Main results**   In Table 2 we show the Top-1 accuracy achieved in 4-bit training using LUQ to quantize the neural gradients to FP4 and combined with a previously suggested method, SAWB (Choi et al., 2018a), to quantize the weights and activations to INT4. We compare our method with Ultra-low (Sun et al., 2020) showing better results in all the models, achieving SOTA in 4-bit training. Moreover, we improve the results by using the proposed SMP (Section 5.1). In Table 3 we show the effect of the proposed fine-tuning, reducing or closing completely the gap from full-precision model. We verified that stochasticity has only a negligible effect on the variance of final performance by running a few different seeds. Additional experiments appear in Appendix A.5.

Table 2: Comparison of 4-bit training of the proposed method LUQ with Ultra-low (Sun et al., 2020) in various vision models with ImageNet dataset, Transformer-base in WMT En-De task dataset and BERT fine-tune in SQUAD dataset. SMP refers to doing two samples of the SR quantization of neural gradients in order to reduce the variance (Section 5.1).

| Model | Baseline | Ultra-low [3] (Sun et al., 2020) | LUQ | LUQ + SMP |
|---|---|---|---|---|
| ResNet-18 | 69.7 % | 68.27% | 69.09% | 69.24 % |
| ResNet-50 | 76.5% | 74.01% | 75.42 % | 75.63 % |
| MobileNet-V2 | 71.9 % | 68.85 % | 69.55 % | 69.7 % |
| ResNext-50 | 77.6 % | N/A | 76.02 % | 76.12 % |
| Transfomer-base | 27.5 (BLEU) | 25.4 | 27.17 | 27.25 |
| BERT fine-tune | 87.03 (F1) | N/A | 85.75 | 85.9 |
| ViT B | 76.2 | N/A | 73.7 % | 74.1 % |

Table 3: Effect of the proposed FNT method (Section 5.2) using FP16 format with different epochs.

| Model | Baseline | LUQ + SMP | +FNT 1 epoch | +FNT 2 epochs | +FNT 3 epochs |
|---|---|---|---|---|---|
| ResNet-18 | 69.7 % | 69.24 % | 69.7 % | - | - |
| ResNet-50 | 76.5 % | 75. 63% | 75.89 % | 76 % | 76.18 % |
| MobileNet-V2 | 71.9 % | 69.7 % | 70.1 % | 70.3 % | 70.3 % |
| ResNext-50 | 77.6 % | 76.12% | 76.25 % | 76.33 % | 76.7 % |

**Overhead of SMP and FNT**   We limit our experiments with the proposed SMP method to only two samples. This is to achieve a similar computational overhead as Ultra-low(Sun et al., 2020), with their suggested two-phase-rounding (TPR) which also generates a duplication for the neural gradient quantization. Additional ablation study of the SMP overhead appears in Appendix A.5. The throughput of a 4-bit training network is approximately 8x in comparison to full precision training (Sun et al., 2020). This means that doing one additional epoch in high precision reduces the throughput by $\sim 8\%$. In comparison, Ultra-low (Sun et al., 2020) does full-training with all the 1x1 convolutions in 8bit, which reduces the throughput by $\sim 50\%$ in comparison to all 4bit training.

---

[3]Recall Ultra-low used a non-standard radix-4 quantization format, that significantly reduces the benefit of using low bit quantization.

**Forward-backward ablations** In Appendix Table 6 we show the Top-1 accuracy in ResNet50 with different quantization schemes. The forward phase (activations + weights) is quantized to INT4 with SAWB (Choi et al., 2018a) and the backward phase (neural gradients) to FP4 with LUQ. As expected, the network is more sensitive to the quantization of the backward phase.

## 7 DISCUSSION

**Conclusions** In this work, we analyze the difference between two rounding schemes: round-to-nearest and stochastic-rounding. We showed that, while the former has lower MSE and works better for the quantization of the forward phase (weights and activations), the latter is an unbiased approximation of the original data and works better for the quantization of the backward phase (specifically, the neural gradients).

Based on these conclusions and previous works (Chmiel et al. (2021)) that showed the optimally of logarithmic quantization, we propose the first method that combined logarithmic quantization with unbiasedness, with the proposed logarithmic unbiased quantizer (LUQ) which quantize the neural gradients to format FP4 [1,3,0]. Combined with a known method for quantizing the weights and activations to INT4 we achieved, without overhead, state-of-the-art in standard format 4-bit training in all the models we examined, e.g., 1.1% error in ResNet50 vs. 2.49% for the previous known SOTA (Sun et al. (2020), which used non-standard format).

Moreover, we suggest two more methods to improve the results, with overhead comparable to Sun et al. (2020). The first reduces the quantization variance, without affecting the unbiasedness of LUQ, by averaging several samples of stochastic neural gradients quantization. The second is a simple method for fine-tuning in high precision for one epoch. Combining all these methods, we were able for the first time to achieve 0.32% error in 4-bit training of ResNet50 ImageNet dataset.

**Reducing the data movement** So far, we focused on resolving the 4-bit GEMM operation bottleneck in DNNs training. It reduces not only the computational resources for the GEMM operations, but also reduces the required memory in DNNs training. However, LUQ, similarly to previous quantization methods (Sun et al., 2020; Choi et al., 2018b;a), requires a statistical measurement of the tensor to define the quantization dynamic range. Specifically, LUQ requires a measurement of the maximum of the neural gradient tensor. Such measurement increases the data movement from and to memory, making this data movement a potential bottleneck in some hardware.

In order to avoid this bottleneck when it is a major issue, we verified that LUQ can be combined with the In-hindsight Fournarakis & Nagel (2021) statistics estimation method, which uses a pre-computed measurement to quantize the current tensor and in parallel extract the current statistics for the next iteration. The maximum estimate in LUQ, $\hat{m}$ is calculated as:

$$\hat{m}^t = (1 - \eta) \cdot \max(|x^{t-1}|) + \eta \cdot \hat{m}^{t-1}, \tag{18}$$

where $\eta$ is the momentum hyperparameter, and $x$ is tensor of neural gradients. In Appendix A.5.5 we show the effectiveness of this statistic estimation, which eliminates this data movement bottleneck, with a negligible change to the accuracy. Though this method can potentially introduce some bias to the quantization process, the bias seems to be negligible (see Appendix Fig. 7).

**Reducing the cost of the scaling operation** In Appendix A.7 we present an optional method to convert the underflow threshold $\alpha$ (Eq. (11)) to a power-of-two, at a small cost to accuracy. This can reduce or eliminate the computational overhead of the multiplication by $\alpha$ in eq. Eq. (12), which can be significant in some cases.

**Multiplication free backpropagation** In this work, we reduce the GEMM bottleneck by combining two different data-types for the forward (INT4) and backward (FP4) passes. Standard GEMM operations with different formats, require casting the operand to a common datatype before the multiplication. The cost of the casting operation can be significant. We notice, that we are dealing with a special case, where one of the operands includes only a mantissa (weights and activations) and the other only an exponent (neural gradients). Our initial analysis (Appendix A.8) shows that this allows, with small hardware changes, reduction of the area of standard GEMM block by $5\times$.

**Accumulation width** A different future direction is to reduce the accumulator width, which is usually kept as FP32. As explained in Appendix A.8, the FP32 accumulator is the most expensive block when training in low bits. Now, after allowing training with 4-bit it is reasonable to think that the accumulator width can be reduced.

## ACKNOWLEDGEMENT

The research of DS was Funded by the European Union (ERC, A-B-C-Deep, 101039436). Views and opinions expressed are however those of the author only and do not necessarily reflect those of the European Union or the European Research Council Executive Agency (ERCEA). Neither the European Union nor the granting authority can be held responsible for them. DS also acknowledges the support of Schmidt Career Advancement Chair in AI.

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
