# OpenReview forum: "Accurate Neural Training with 4-bit Matrix Multiplications at Standard Formats"
_ICLR.cc/2023/Conference — ICLR 2023 poster_

### Official Review · Reviewer_8NA6 · 2022-10-21

**Confidence:** 4
**Correctness:** 3
**Technical Novelty And Significance:** 2
**Empirical Novelty And Significance:** 2
**Recommendation:** 6

**Clarity, Quality, Novelty And Reproducibility:**

The experimental results, and crucially, the implementation details in the Appendix look good.

Finally, there are some typos and grammatical errors throughout the paper. I urge the authors to perform a spell check.


**Strength And Weaknesses:**

While this paper is interesting and presents promising results, I did have a few questions listed hereafter:
Section 3: In the calculation of rounding variance and MSE, why is the input x considered deterministic?

In Section 3's conclusion, the authors claim that the quantization MSE has to be minimized in the forward pass, which is a reasonable claim. Some works have shown how to provably do that for INT4 [1]. Can this work be compared to the proposed technique?
[1] Sakr, Charbel, et al. "Optimal Clipping and Magnitude-aware Differentiation for Improved Quantization-aware Training." International Conference on Machine Learning. PMLR, 2022.


**Summary Of The Paper:**

This paper proposes techniques to quantize the gradients in the back-propagation algorithm down to 4 bits. To do so, the authors propose the LUQ format, as well as bias removal and variance reduction techniques. In order to achieve SOTA accuracy, the method requires an additional stage of high precision fine-tuning.

**Summary Of The Review:**

The paper is interesting, tackles an important problem, and presents promising results. I raised some questions and hope to receive responses from the authors.

---

> ### Author Response · Authors · 2022-11-10
> **Response to reviewer 8NA6**
>
> $\textbf{Q1}$: "In order to achieve SOTA accuracy, the method requires an additional stage of high precision fine-tuning."
>
> $\textbf{A1}$: The proposed LUQ already achieved SOTA in 4-bit training without any fine-tuning as presented in Table 2. The fine-tuning method just improve the results, with some cost as discussed in the paragraph "Overhead of SMP and FNT" in section 6.
>
> $\textbf{Q2}$: " Section 3: In the calculation of rounding variance and MSE, why is the input x considered deterministic?"
>
> $\textbf{A2}$: Yes, the derivation in section 3 is for a known deterministic tensor x, since the randomness is only on the quantizer. This is quite general, since it means our results (e.g., equation 6) also apply if x is a random variable (for any probability distribution), as we explain in footnote 1 in page 4.
>
> $\textbf{Q3}$: "In Section 3's conclusion, the authors claim that the quantization MSE has to be minimized in the forward pass, which is a reasonable claim. Some works have shown how to provably do that for INT4 [1]. Can this work be compared to the proposed technique? [1] Sakr, Charbel, et al. "Optimal Clipping and Magnitude-aware Differentiation for Improved Quantization-aware Training." International Conference on Machine Learning. PMLR, 2022."
>
> $\textbf{A3}$: We thank the reviewer for presenting us this work, we added it to the related works section in the updated version. The proposed LUQ
> aim to quantize only the neural gradients and we combined it with a known method (SAWB [A]) to quantize the forward phase (weight and activations).  During our research we tried additional known INT4 methods for the forward phase, including PACT[B], which learns the clipping value. We believe the reviewer's proposed technique can be similarly combined with LUQ. As the reviewer suggests, since it minimizes the MSE, maybe it can improve the 4-bit training results in the combination with LUQ. We definitely plan to check this direction.
>
>
> $\textbf{Q4}$: Finally, there are some typos and grammatical errors throughout the paper. I urge the authors to perform a spell check.
>
>
> $\textbf{A4}$: We thank the reviewer for his comment. We applied another pass with a spell checker in the updated version.
>
>
>
>
> ===============================
>
> [A] Jungwook Choi, P. Chuang, Zhuo Wang, Swagath Venkataramani, V. Srinivasan, and K. Gopalakrish-
> nan. Bridging the accuracy gap for 2-bit quantized neural networks (qnn). ArXiv, abs/1807.06964,
> 2018a
>
> [B] Jungwook Choi, Zhuo Wang, Swagath Venkataramani, P. Chuang, V. Srinivasan, and K. Gopalakrish-
> nan. Pact: Parameterized clipping activation for quantized neural networks. ArXiv, abs/1805.06085,
> 2018b.

---

### Official Review · Reviewer_8T7H · 2022-10-24

**Confidence:** 4
**Correctness:** 3
**Technical Novelty And Significance:** 3
**Empirical Novelty And Significance:** 2
**Recommendation:** 6

**Clarity, Quality, Novelty And Reproducibility:**

The overall writing is relatively easy to follow. It would help improve the clarify and quality by resolving the a few technical clarifications discussed in strength and weakness.

**Strength And Weaknesses:**

Strength

1. The effort and design towards the simplicity is impressive. The key of the practical use of low precision training is the simplicity which could be efficiently implemented and embraced by the HW/SW practitioners. Previous methods needs very special hardware design or algorithmic pieces that could trigger more compute overhead than the savings of going to 4-bit. The LUQ method uses standard Radix-2 and just standard way of stochastic rounding. By choosing the right rounding method for forward and backward pass. This simple method could achieve stronger accuracy than previous method.

2. The two optional methods to improve 4-bit accuracy is practical and potentially lead to impact on future hardware designs to accommodate these considerations.


Weakness

Some of the technical and writing aspects need some more clarity to be fully evaluated.

1. Around Equ 13 and 14, the authors discussed a method to correct the inherent bias in directly rounding the exponent. Is the intention to discuss an exact or approximated approach to correct the bias yet still preserve the hardware efficiency of directly rounding the exponents? I ask this question mostly because that converting back to linear space from logarithmic space and then do the stochastic rounding might trigger additional computation overhead. If this discussion intends to tell how rounding on exponent can achieve bias correction, further elaborations are needed to clarify how the bias correction is achieved.

2. The value of \alpha in Equ 18 determines the exact interval end point values of LUQ. What is the way of determining \alpha in practice? Seems from the beginning of page 6, \alpha is determined by the gradient value range. Does this need dataset / model specific values in practice? If yes, a discussion on the method to determination in the main text or appendix would would be useful. This is important for understanding the practical overhead of enabling the 4-bit training using LUQ.

3. Around Equ 15, the authors claimed the quantized gradient is unbiased. It seems this is under the assumption that the activation is unbiased. If this is the case the assumption should be elaborated to avoid conflations with the fact that LUQ suggest biased forward activation quantization.


**Summary Of The Paper:**

This paper proposed a 4-bit quantization method (LUQ) for matrix multiplication in training deep learning models. This method uses 4-bit integer quantization for weight and activation quantization, and use the proposed logarithmic unbiased quantization (LUQ) for gradient back  propagation. The major contributions are the following:

1. A study showing that at the 4-bit quantization level for matmul, stochastic quantization is preferred for backward pass while the nearest rounding presents better accuracy when applied in forward pass.

2. The first training pipeline with fully 4-bit matmul using standard format. With the LUQ design, this pipeline does not require special techniques like Radix-4 or double pass rounding [Sun et al. 2020]. Yet, it gives the SoTA 4-bit training accuracy with small accuracy degradations to single precision training on several NLP and vision tasks.

3. Two practical techniques to further improve the accuracy of 4-bit training 1) Variance reduction with resampling for stochastic rounding 2) high precision continuous training.


Reference:
[Sun et al. 2020] Ultra-Low Precision 4-bit Training of Deep Neural Networks

**Summary Of The Review:**

Overall I recommended marginally-above-acceptance. I think the LUQ method provides ways to achieve 4-bit matmul in forward and backward pass using standard hardware and software techniques, which opens the door towards system and hardware practical 4-bit training. The writing and technical discussion could be further improved by resolving clarity comments above. I would highly recommend some centralized discussion about practical way to determining the threshold (\alpha) in LUQ method. If there is a simple yet effective way to achieve this, the impact will be much stronger.

---

> ### Author Response · Authors · 2022-11-10
> **Response to reviewer 8T7H**
>
> $\textbf{Q1}$: "Around Equ 13 and 14, the authors discussed a method to correct the inherent bias in directly rounding the exponent. Is the intention to discuss an exact or approximated approach to correct the bias yet still preserve the hardware efficiency of directly rounding the exponents? I ask this question mostly because that converting back to linear space from logarithmic space and then do the stochastic rounding might trigger additional computation overhead. If this discussion intends to tell how rounding on exponent can achieve bias correction, further elaborations are needed to clarify how the bias correction is achieved."
>
> $\textbf{A1}$: We thank the reviewer for his question. The intention of this section is to show that we are able to apply stochastic rounding directly (and exactly) on the exponent, without the need to go back to linear scale ---- which would trigger additional computation overhead, as mentioned by the reviewer. Specifically, since stochastic-rounding requires a round-to-nearest operation, we showed that standard round-to-nearest on the exponent is biased, and we suggested RDNP to fix this bias. In LUQ, in order to build the first logarithmic unbiased quantizer, we apply RDNP on $x + \varepsilon$ where $\varepsilon$ is a random noise. We clarify this point in the updated version.
>
> $\textbf{Q2}$: "The value of $\alpha$ in Equ 18 determines the exact interval end point values of LUQ. What is the way of determining $\alpha$ in practice? Seems from the beginning of page 6, $\alpha$ is determined by the gradient value range. Does this need dataset / model specific values in practice? If yes, a discussion on the method to determination in the main text or appendix would would be useful. This is important for understanding the practical overhead of enabling the 4-bit training using LUQ."
>
>
> $\textbf{A2}$: We believe the reviewer refers to equation 11 (and not 18, which does not include $\alpha$).  The value of $\alpha$ in the vanilla version of LUQ is calculated on-the-fly during training according to the range of the tensor: i.e., in each iteration we calculate the maximum of the neural gradient and calculate $\alpha$ as presented in equation 11. This value changes during training and depends on the model and dataset. We agree with the reviewer that measuring this value may have some overhead, and this is precisely the reason we presented in the discussion section (and related appendix sections) two additional methods to reduce the overhead: (1) In the paragraph "Reducing the data movement" we show it is possible to use the maximum of previous iterations instead of current maximum. This way, we do not need to wait to go over all tensor to calculate the maximum to continue the training. This will reduce dramatically the data movement, and as we showed, minimally affects accuracy. (2) In the paragraph "Reducing the cost of the scaling operation" we showed we can approximate the maximum to the close power-of-two value in order to reduce the overhead of the calculation of $\alpha$. Moreover, we present  the combination of both methods (Table 8 in the appendix). We added a clarification on this near equation 11
>
>
> $\textbf{Q3}$: "Around Equ 15, the authors claimed the quantized gradient is unbiased. It seems this is under the assumption that the activation is unbiased. If this is the case the assumption should be elaborated to avoid conflations with the fact that LUQ suggest biased forward activation quantization"
>
>
> $\textbf{A3}$: Perhaps there is some misunderstanding. Around that equation we only say that the quantization method of the neural gradient tensor (i.e., both the Q and T operators we defined in that section) is unbiased. More generally, as we discuss in section 3, quantizing the activations and weights introduces an unavoidable bias, already in the forward pass (around equation 9). So the goal of LUQ is not to introduce any additional bias when quantizing the neural gradients (the forward pass is quantized using an existing method SAWB, which is not part of LUQ).

---

### Official Review · Reviewer_Qvzo · 2022-10-24

**Confidence:** 4
**Correctness:** 3
**Technical Novelty And Significance:** 3
**Empirical Novelty And Significance:** 3
**Recommendation:** 8

**Clarity, Quality, Novelty And Reproducibility:**

Clarity: The paper is mostly clearly written, but it might require slightly more training details. For example, in the training phase, is just SAWB applied without any modification? (Training from scratch is more challenging than training from converged full-precision model used in commly quantization-aware training setting, and the result looks somewhat too good to me.) Is any other tricks such as chunk-based accumulation adopted?

Quality: The technical is good.

Novelty: The novelty is somewhat thin, as pointed out before.

Reproducibility: I didn't run the code, but the code looks to be reproducible to me.

**Strength And Weaknesses:**

Strength:
- The proposed approach is simple.
- The proposed approach achieves good accuracy, the result is impressive.
- Training with low-bit arthemetic is a major problem. The paper can be significant.
- The experiments cover a wider range of applications than other papers in the field.

Weaknesses:
- The novelty is somewhat thin: Until the second half of page 5, the paper is mostly presenting existing backgrounds. The novelty mainly falls in Sec. 4. But the LUQ itself is rather straightforward to design, once the goal of designing logarithmic and unbiased quantizer is clear. The approaches in Sec. 5 are also rather standard and to some extent explored in previous literature. I'd say the main contribution of this paper is   showing that such a simple combination of existing techniques is sufficient to achieve (surpringly good) accuracy, rather than proposing novel techniques.

**Summary Of The Paper:**

The paper proposes a method to train neural networks with 4-bit matrix multiplications. The authors propose logarithm unbiased quantization (LUQ), which combines stochastic rounding and logarithm quantization, both of which are known to be required for extreme low-bit training. The proposed training algorithm involves 4-bit deterministic integer quantization for weight and activation and 4-bit LUQ for gradients. The authors further propose some optional methods to improve the accuracy, including averaging the quantized gradient (SMP) and high-precision fine-tuning (FNT). On various computer vision and natural language processing benchmarks, the proposed approach can train with 4-bit numerical formats, with superior accuracy than previous FP4 training.

**Summary Of The Review:**

Though the novelty is somewhat thin. I still think this is a good paper due to
1. Its simplicity and effectiveness;
2. Its reproducibility. At least the official code is provided, which I suspect to be the first open-sourced recipe for 4-bit training.

---

> ### Author Response · Authors · 2022-11-10
> **Response to reviewer Qvzo**
>
> $\textbf{Q1}$: "The novelty is somewhat thin: Until the second half of page 5, the paper is mostly presenting existing backgrounds. The novelty mainly falls in Sec. 4. But the LUQ itself is rather straightforward to design, once the goal of designing logarithmic and unbiased quantizer is clear. The approaches in Sec. 5 are also rather standard and to some extent explored in previous literature. I'd say the main contribution of this paper is showing that such a simple combination of existing techniques is sufficient to achieve (surprisingly good) accuracy, rather than proposing novel techniques."
>
> $\textbf{A1}$: Indeed, our main technical novelty is LUQ, which design is straightforward, once it is clear our goal is to find a logarithmic and unbiased quantizer. However, the fact this design was not done until now, suggests this goal was not sufficiently clear in the literature. This is exactly why we wrote the "pedagogical" section 3, to motivate and formulate this important goal. The main empirical novelty in this paper is the surprising observation that, by simply taking this goal seriously, we can obtain SOTA results (and with a simple and efficient method). Since we did not want to detract the focus from these central findings, we moved other parts with technical novelty (but outside the scope of these main contributions) to the discussion and related appendices (e.g., Section A.8, "Multiplication free backpropagation"). We hope this clarifies our view regarding the contributions of this paper, and how it was written.
>
>
> $\textbf{Q2}$: "in the training phase, is just SAWB applied without any modification? (Training from scratch is more challenging than training from converged full-precision model used in commly quantization-aware training setting, and the result looks somewhat too good to me.) Is any other tricks such as chunk-based accumulation adopted?"
>
>
> $\textbf{A2}$: In the training phase we use standard SAWB [A], i.e. for each weight $W$ the clipping value is $S = c_1 \cdot \sqrt{\mathbb{E}(W^2)} - c_2 \cdot \mathbb{E}(|W|)$ where $c_1$ and $c_2$ are coefficient predetermined from linear regression over 6 known distributions (For 4-bit $c_1 = 12.1$, $c_2 = 12.2$. The quantization is per layer and not chunk-based. We agree with the reviewer that training from scratch is more challenging that quantization-aware-training. This is the reason we tried various known methods for INT4 quantization of the forward phase, including methods which learn the clipping value such as PACT [B]. We found that SAWB [B], which determines the clipping value according to the statistics works better, for example for ResNet50, training from scratch with SAWB for the weights and activation (neural gradients in full precision) we achieved 76.35\% , while using PACT we achieved 75.7\%.
>
>
> ===============================
>
> [A] Jungwook Choi, P. Chuang, Zhuo Wang, Swagath Venkataramani, V. Srinivasan, and K. Gopalakrish-
> nan. Bridging the accuracy gap for 2-bit quantized neural networks (qnn). ArXiv, abs/1807.06964,
> 2018a
>
> [B] Jungwook Choi, Zhuo Wang, Swagath Venkataramani, P. Chuang, V. Srinivasan, and K. Gopalakrish-
> nan. Pact: Parameterized clipping activation for quantized neural networks. ArXiv, abs/1805.06085,
> 2018b.

---

> > ### Comment · Reviewer_Qvzo · 2022-12-07
> > **Thanks for the response**
> >
> > Thanks for the clarifications. The response looks good to me.

---

### Public Comment · ~Haocheng_Xi1 · 2022-11-07
**Hope to update the code to reproduce the result**

Hello, I am very interested in your work, which has very good performance. However, I notice that your code is missing some files, so I can not reproduce your result using LUQ in the backward pass and SAWB in the forward pass.(For example, in vision/models/modules/se.py it import .activation, but it does not include such a file) Can you upload the code that can reproduce your result on the Imagenet? I am very grateful for this. (I believe that the method works since I implement it by myself and get relatively good result, but it is better to look at the original code :)

---

> ### Author Response · Authors · 2022-11-10
> **Response to Haocheng Xi**
>
> Thank you for your comment.
> The code for the pass backward of LUQ appears in lines 97-121 in the file LUQ.py. The code of the forward pass of SAWB appears in line 69-79 in the same file. We uploaded the file .activation.py, however it is not required to reproduce the experiments. Please let us know if there are any other issues.

---

### Author Response · Authors · 2022-11-10
**General comment**

We thank all the reviewers for their detailed feedback, and for expressing positive opinions: Reviewer Qvzo --- $\textit{"Its simplicity and effectiveness"}$; Reviewer 8T7H --- $\textit{"This simple method could achieve stronger accuracy than previous method."}$; and Reviewer 8NA6 --- $\textit{"The paper is interesting, tackles an important problem, and presents promising results"}$. We uploaded a new revision of the paper and supplementary material to address all remarks. All the changes are marked in red. For additional details, see the answer to each reviewer's concerns. Please let us know if there are any additional comments. If we addressed all concerns, we kindly ask the reviewers to increase their scores.

---

### Decision · Program_Chairs · 2023-01-20

**Decision:**

Accept: poster

**Justification For Why Not Higher Score:**

The scope is limited to model quantization, the improvement is real but not extraordinary. There are some minor flaws which make it a "not totally seamless method" to deploy / change our training loops with.

**Justification For Why Not Lower Score:**

This is a good paper, giving it more visibility may help follow-up research. The flaws are not deal-breakers.

**Metareview: Summary, Strengths And Weaknesses:**

This paper shows how to perform 4-bit quantization training for deep learning models. The method uses 4-bit integer quantization for weight *and* activation quantization, and use logarithmic unbiased quantization (LUQ) and stochastic rounding for gradient back propagation. LUQ and stochastic rounding are not new, and the main novelty of the paper is to combine them and perform experimental validations. The methods improves over the state of the art for full training quantization by using a small high precision fine-tuning step, which can be seem as a remaining minor flaw. The paper is clear, the experiments (ResNets and Transformers, ImageNet/WMT/Squad datasets and tasks) support the claims adequately.

**Note From Pc:**

if the above contains the word "oral" or "spotlight" please see: "oral" presentation means -> notable-top-5% and "spotlight" means -> notable-top-25%. As stated in our emails, we are disassociating presentation type from AC recommendations

**Summary Of Ac-Reviewer Meeting:**

N/A